# Efficacy of the Feliway® Classic Diffuser in reducing undesirable scratching in cats: A randomised, triple-blind, placebo-controlled study

**Joana Soares Pereira**[1], **Yasemin Salgirli Demirbas**[2], **Laurianne Meppiel**[3], **Sarah Endersby**[3], **Gonçalo da Graça Pereira**[1], **Xavier De Jaeger**[3]*

**1** Egas Moniz Center for Interdisciplinary Research (CiiEM), Egas Moniz School of Health and Science, Caparica, Almada, Portugal, **2** Department of Physiology, Faculty of Veterinary Medicine, Ankara University, Ankara, Turkey, **3** Ceva Santé Animale, Libourne, France

* xavier.de-jaeger@ceva.com

**Data Availability Statement:** All relevant data are within the manuscript and its Supporting information files.

## Abstract

Scratching the environment is a natural behaviour that cats use for communication and physical maintenance purposes, however when it is carried out on household furniture it is considered unacceptable for some owners and even grounds for relinquishment of cats. The objective of this study was to investigate the efficacy of FELIWAY® Classic Diffuser in reducing undesirable scratching (scratching vertical surfaces indoors other than the scratching post) in cats. A 28 day, randomised, triple-blind, placebo-controlled study with a total of 1060 caregiver-cat dyads was conducted. The study contained two groups: the Pheromone Group consisted of caregivers who were given a pheromone diffuser (n = 546) and the Placebo Group consisted of caregivers who were given a placebo diffuser (n = 514). A questionnaire with three subsections was distributed online. The first section, completed by the respondents at day 0, inquired about the cats' daily routines, social and physical environments, behaviour, temperament, and emotional states. The second section filled on day 0, 7, 14, and 28, assessed the Frequency and the Intensity of the undesirable scratching problem and the effectiveness of the product. The last section, filled on the 28th day of the product application, related to the caregivers' opinions about the product and overall outcome. After 28 days the scratching Frequency reduced for 83.5% of the cats in the Pheromone Group and 68.5% for the Placebo Group (p<0.0001). The Intensity was significantly different between treatment groups at D7 (p = 0.0170), at D14 (p = 0.0189) and at D28 (p<0.001). The reduction of the Global Index Score, which was calculated by multiplying the Intensity with the Frequency, was significantly higher for the Pheromone Group (p<0.001). This study provides direct evidence that the use of FELIWAY® Classic diffuser significantly reduces the Frequency, Intensity and the Global Index Score of undesirable scratching.

**Funding:** This study was funded by Ceva Santé Animale who provided the diffusers and was implicated in all the step of this clinical trial.

**Competing interests:** LM, SE and XDJ are employees of Ceva Santé Animale. JSP, YS, GGP authors received fees from Ceva Santé Animale for their contribution on the study design, interpretation of results and manuscript writing. This does not alter our adherence to PLOS ONE policies on sharing data and materials.

## Introduction

Cats have a clear behavioural need to scratch their environment, both horizontally and vertically. They exhibit this behaviour for a number of purposes, such as maintaining their front claws, stretching, exercising or visual and chemical marking [1, 2]. When scratching and rubbing their heads, cats deposit secretions produced by the sebaceous glands, located in their interdigital and facial areas, that carry semiochemicals [3, 4]. These semiochemicals convey species-specific sexual, territorial and spatial information messages that are received by a member of the same species (including the individual who delivered it) and trigger a behavioural response [5]. When scratching is observed repeatedly inside the house it could indicate the cat is not feeling safe in that environment [1, 6]. Regardless of the underlying motivation and stress, scratching is a normal feline behaviour. Despite this, a considerable number of caregivers report it as a behavioural problem if it causes damage to household furniture (chairs, sofas, walls etc.) [7] and some have even cited it as a cause for relinquishment [8, 9].

Several strategies are used to reduce the frequency and mitigate the undesirable consequences of scratching, namely: providing suitable scratching devices, the use of semiochemicals to redirect the behaviour to an acceptable scratching device, regular trimming of the nails, applying plastic nail covers, chemical deterrents or fear inducing stimuli systems, keeping the cat strictly outdoors, and surgical declawing (onychectomy) [1, 10–12]. Despite the fact that it is forbidden in a lot of countries such as Germany, declawing is performed in 20 to 45% of cats in different states within the US [10, 13]. This procedure is strongly opposed by the American Association of Feline Practitioners (AAFP), the American Animal Hospital Association (AAHA) and the Canadian Veterinary Medical Association (CVMA) as it can cause intense acute pain and potentially lead to chronic physical discomfort and restrict the exhibition of natural behaviour. Plastic nail covers and chemical deterrents are also controversial options as they impede this natural behaviour, which could have lasting negative implications on the welfare of cats [10, 14–19]. Even the use of a scratcher may not always meet the cat's needs [20], as evidenced by Beck and colleagues (2018) who found that of the cats who used the provided scratchers, a high number would still scratch furniture [10]. In order to meet the general preferences of cats, a scratching post should: be vertical, have a sisal rope as substrate, be more than 0.9m (3 feet) high with a base of 0.3 to 0.9m (1 to 3 feet), and two or more levels. Kittens seem to prefer softer materials as substrate such as cardboard and S-shaped scratchers. It is recommended for the scratching post to be placed near the cat's sleeping areas, the borders of the cat's territory (doors, windows. . .), and areas where the cat has previously scratched on the household furniture [1, 6].

In the United States, the United Kingdom and others European countries, the number of cats kept as pets has been on the rise [21–23]. A deeper knowledge of the scratching needs and preferences of cats, as well as an understanding on how to encourage the use of scratching posts is needed. In fact, it could help prevent unnecessary surgeries and improve welfare in this growing pet cat population [6–8, 12, 24]

Under stressful conditions, the frequency and severity of scratching behaviour in cats will increase since it functions as a marking response [4, 25]. Although the link between stress and scratching has been established, studies exploring the effect of reducing overall stress on scratch marking behaviour are still lacking. It is well known that olfactory communication is essential for cats to determine the safety of their surroundings; the brain areas controlling stress response and the olfactory system are intimately connected, and olfactory methods offer special chances to change stress and associated behaviour problems in cats [1, 6, 26]. The synthetic analogues of semiochemicals have a significant impact on the behaviour and wellbeing of cats, for example the F3 fraction of the Feline Facial Pheromone (FFP) contributes to the

reduction of stress induced behaviours, increases calm behaviours and reduces urine marking [4, 5]. The synthetic analogue of the F3 fraction is also reported as a potential way to decrease scratching behaviour [3], however to the authors knowledge there are no double or triple-blinded, placebo-controlled studies regarding the efficacy of the F3 fraction alone on the reduction of undesired scratching.

The aim of this study was to investigate the impact of the use of FELIWAY® Classic Diffuser in reducing undesirable scratching in cats.

## Materials and methods

### Study design

The authors designed a randomised, triple-blinded, placebo-controlled, home-based caregiver study that was carried out in France. The participants were part of a panel of cat caregivers and were contacted by e-mail. The selection questionnaire was sent to them and it was asked if they wanted to be part of a study.

Two identical diffusers provided by Ceva® Santé Animale were used: one containing the synthetic version of the F3 fraction of the FFP–FELIWAY® Classic, and another containing a placebo solution (the solvent without the F3 Fraction). The two products were strictly identical and identified with a four-digit number. Participants were divided in two groups: those who were given the pheromone diffuser (Pheromone Group, n = 546) and those who were given the placebo diffuser (Placebo Group, n = 514). The allocation of the product ID and the group was randomized by block of 4, and the product was distributed in increasing order to the caregivers. The duration of the study was 28 days.

### Data collection

For this study, the questionnaires were distributed through a web application and the caregivers were informed when each questionnaire was available via email and the end the study a gift card was deliver to the caregiver. After 24h a reminder was sent by email and SMS to the caregiver and they had 24h to fill out the questionnaire (see Fig 1).

A questionnaire with four subsections was designed:

- the first section inquired about owner and cat's demographic information;

- the second section was completed by the respondents at the beginning of the study and inquired about the cat's daily routines, social and physical environments, behaviour, temperament, and emotional states. This included questions about type of food, body condition, owners' opinion on the cat's temperament and personality, time spent playing, reaction to visits/new people, frequently seen body postures, other behavioural problems, scratching post information etc;

- the third section included questions to assess the Frequency and the Intensity of the scratching problem and the effectiveness of the product on a weekly basis (see supplementary files). In this section owners were also inquired about the cat's body postures. Owners were also asked to report any changes and adverse effects seen on the week before. This was filled by the participants on day 0, 7, 14, and 28 of the study.

- the last section, which included four questions related to the respondents' opinions about the product and the overall behavioural outcome, was delivered the 28th day of the treatment course.

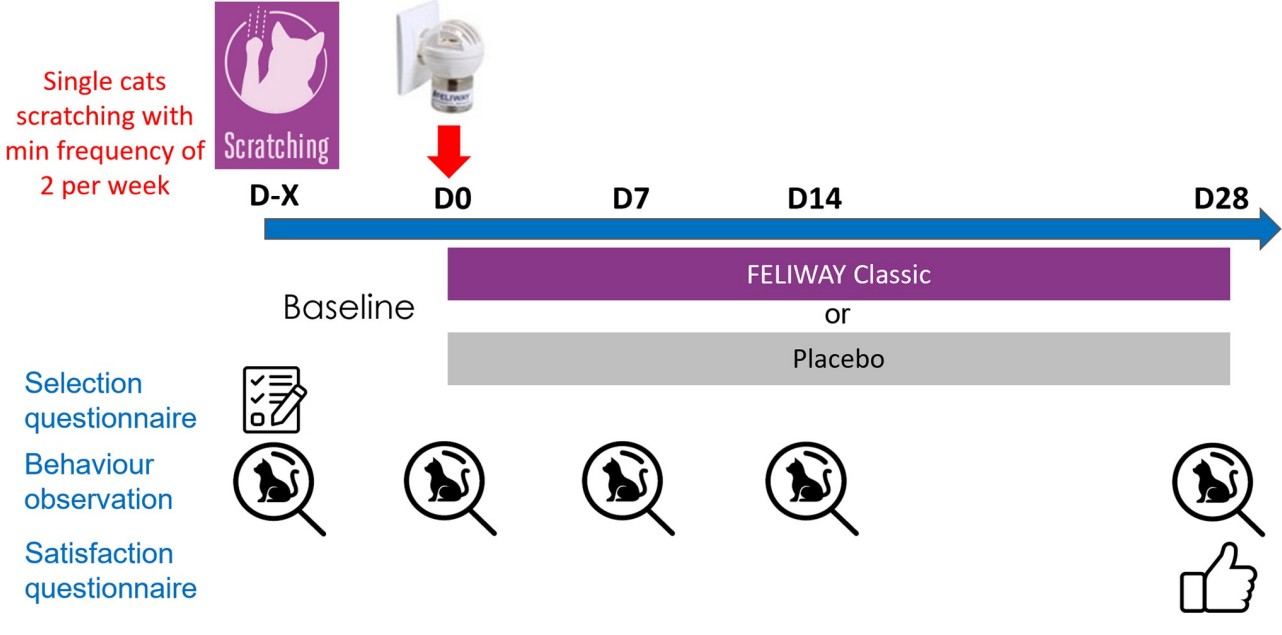

**Fig 1. Scheme of study protocol.**

To be selected, the caregiver needed to be over 18 years old and to have signed their consent. Undesired scratching was defined as scratching on vertical surfaces indoors other than the scratching post, for example the sofa, carpet, curtains, or furniture. Only single cat households with undesirable scratching behavioural reported by the caregiver for at least one month, with a minimal frequency on the scale of two per week (indoor scratching), were included.

To evaluate undesirable scratching Frequency, the caregivers were asked to consider any event directly observed or deduced from new damage observed during the previous week, and to report it on a semi-quantitative scale with 7 degrees (0 = Never, 1 = Once a week, 2 = Twice a week, 3 = Every 2 days, 4 = Almost every day, 5 = Every day, once or twice a day, 6 = Every day, more than twice a day) [27]. Caregivers were asked to report the Intensity of scratching on a visual analogue scale from 1–10 (with intervals of 0.1), considering the duration of the scratching event and/or the extent of the damage. The owner could not rate the Intensity at 0 because it was considered that if scratching was present (Frequency ≠ never) this event should have a minimal intensity. This also avoided bias for the Global Index Score by removing the potential of a 0 value for the index score despite some frequency being recorded.

A Global Index Score of scratching was calculated by multiplying the Intensity by the Frequency, resulting in a score ranging from 0 to 60.

A "Disturbing" value was recorded every week by asking the owners (on a visual analogue scale from 0–10) to what extent did they consider this scratching problem was disturbing for them and their household over the last 7 days. At D28, owners were asked to provide their overall assessment of the scratching problem over the 28 days (worse (higher Frequency or Intensity) / worse (new scratching place) / Didn't change at all / Changed a little (lower Intensity only) / Changed a little (lower Frequency only) / Changed a lot (both Frequency and Intensity) / Changed completely (stopped scratching)).

The animal study was reviewed and approved by Ceva Santé Animale Committee (ref CFAEC-2022-08). Written informed consent was obtained from the owners for the participation of their animals in the study.

## Exclusion criteria

Eleven exclusion criteria were chosen to provide survey responses that accurately reflect effectiveness of the pheromone product (Table 1). Meeting any of the exclusion criteria would result in a participant being removed from the study.

## Statistical analysis

Based on internal data collected by Ceva Santé Animale in France and the USA, and analyzing absolute change of Index score between 0 and 14 days, probabilities of success (to have a significant p-value for the product with a regression model) were calculated with simulated databases to estimate the number of cases required. Considering the worst case, the probability of having a significant product effect was 90% with 1000 cats.

Data was analyzed using SAS v9.4 (SAS Institute) and R software (v4.2.2). The level of significance was set at $p < 0.05$. Undesirable scratching was assessed through three components: the Frequency of the behaviour, the Intensity of the behaviour and the Index Score that represents a global evaluation of the behaviour. The Global Index Score was calculated by multiplying the Frequency by the Intensity of the undesirable scratching behaviour. Due to the randomization process any difference at Baseline must have occurred by chance, so it is not

**Table 1. List of exclusion criteria.**

| Exclusion Criteria | Description of Criteria | Rationale for Exclusion |
|---|---|---|
| Cat age | Cats < 6 months and cats > 11 years | Age related changes in frequency and intensity of scratching |
| Outdoor access | Cats spending most of their time outdoors | Inability to control external stressors |
| Environmental Changes | • Cats returning home less than 15 days prior to the study after hospitalization, cattery or shelter stay<br>• Major environmental change (e.g. moving to a new house, new pet, major change of furniture, new owner schedule) | Possible increase in cat perceived stress |
| Household composition | Multicat household | Inability to identify the source of undesired scratching |
| Cat's health issues (including behavioural) | • Cats with osteoarthritis diagnosed by a veterinarian, recovering from orthopedic surgery for less than 6 weeks or any other health problem/intervention that resulted in the cat being prostrated<br>• Cats under the care of a vet or a behaviourist because of the undesired scratching problem | • Disease and pain related changes in Frequency and Intensity of scratching<br>• Inability to evaluate the single effect of the product |
| Other products | Current or recent (in the past 6 months) use of other pheromone products or calming products (nutraceutical, pharmaceutical, environmental product...) | Inability to evaluate the single effect of the product |
| Working Area of Caregivers | • Caregivers belonging to a research panel dedicated to a specific brand or a specific product manufacturer<br>• Caregiver working in one of the following areas: pharmaceutical industry, a veterinary practice or clinic, a pet store or shop, manufacture or distribution of pet food or pet care products, survey / market research institute | Possible bias while answering questions |
| **Data** point missing | If one questionnaire was not answered whatever the reason. | If a data point is missing the calculation of the change would not be possible |
| Observation not possible | Caregiver that was not able to observe the cat for at least 5 days during the week and at least 4 hours each day | Difficulty in observing behavioural changes |

appropriate to report the results of significance tests comparing the two groups at baseline [28].

For quantitative parameters, the number of animals, the mean and standard deviation were calculated. For qualitative parameters, Frequency distributions and number of animals were detailed.

For scratching Frequency, since the data are ordinal, medians were presented and a Wilcoxon test was performed. A cumulative Link model for ordinal regression was also performed to determine the probabilities of being in each scratching Frequency category according to days and product group. To limit the number categories, we regrouped them two by two for this analysis (Never (0 per week) / once or twice a week (1 or 2 per week) / every two days and almost every day (3 to 6 per week) / every day and more (7 and more per week)). The Frequency at baseline, treatment group, day and interaction between day and treatment were added to the model.

A chi-squared test was used to compare the percentage of cats with more or less than 50% reduction of scratching for Frequency, Intensity and Global Index Score.

For both Intensity and Global Index Score, the same analyses were performed: as these two parameters followed a normal distribution, their change from baseline was analyzed using a longitudinal mixed model with repeated measures (MIXED procedure of SAS). Visit day, baseline value and treatment group were fixed effects. Cat age, lifestyle, presence of behavior problem, emotional health, time spent at home, other pets at home, owner's family situation (see Table 2), location of the scratching and number of the scratching post were tested as fixed effect and kept if necessary, using Akaike's information criterion. The animal was tested as a random effect.

Efficacy data was summarized per behaviour observation day (D7, D14 and D28). Observation days (D7, D14, and D28) and baseline (D0) values were fixed effects, and the cat was considered as a random effect. 95% confidence intervals and p-values were provided to analyze the significance of the mean generalized least square means estimates.

For all the models, the covariables presented in Table 2 were tested as fixed effects and kept if necessary to fit the best model (determined by Akaike's information criteria), the interactions between the covariables and the treatment group were tested and kept if significant to fit the best model.

## Results

### Population definition

For this study 1415 cats were recruited. A total of 355 were excluded (see details in Table 3) which resulted in 1060 cats. The Pheromone Group included 546 cats and the Placebo Group included 514 cats in the final analysis.

Both groups had similar characteristics in terms of the number of male / female cats, respectively 50.7% and 49.2% for the Placebo Group and 46.7% and 53.3% for the Pheromone Group (Fig 2A). In both groups the mean Frequency, the mean Intensity and the mean scratching Index were also comparable at baseline (Fig 2B), others baseline characteristics were providing as supporting information (S1 and S2 Tables in S3 File).

The estimation of the scratching Frequency during the selection, based on the caregiver's memory, was similar for both groups with a median of around 4, meaning approximately "Almost every day". Those values were congruent to those recorded at Baseline (D0) and, again, similar between groups (see Fig 2C). The cat personality profile was evaluated through a series of questions (YES/NO) on different personality traits, and again no clear difference appeared between the two groups. It was determined that the randomization was successful in

**Table 2. Covariables tested as fixed effects.**

| | |
|---|---|
| **Cat age** | • 7 months-2 years<br>• 3–6 years<br>• 7–10 years) |
| **Lifestyle of the cat** | • lives exclusively indoors<br>• spends an average of 2 to 4 hours outdoors<br>• spends and average of 4 to 6 hours outdoors<br>• spends in average more than 6 hours outdoors |
| **Presence of behaviour problem** | • no<br>• yes if elimination, biting or activity at night |
| **Emotional health** | picture based |
| **Working time organization** | • I spend most of my days at home (telecommuting, at home. . .)<br>• I spend 1 day away and 4 days at home<br>• I spend 2 or 3 days at home and 2 or 3 days away<br>• I spend 1 day at home and 4 days away<br>• I spend all my days away from home |
| **Other pets at home** | • Dog<br>• Other<br>• No |
| **Caregiver's family situation** | • Single with no children at home / Single with children at home / Married or cohabiting with no children at home / Married or cohabiting with children at home / Prefer not to answer |
| **Location of the scratching** | • The same room as the behavior problem happens<br>• A different room from where the behaviour problem happens |
| **Number of the scratching post** | • 1<br>• More than one |

balancing the cats in the different groups, considering the similarities between the two groups at baseline.

## Efficacy of the synthetic F3 Feline Facial Pheromone fraction on the reduction of scratching

Due to the non-continuous aspect of the scratching Frequency, the medians of the scratching Frequency are presented in Fig 3A. A quantitative test of Wilcoxon was performed and an ordinal regression model was used to calculate the probabilities to be in each category (see S1 Fig). Considering the 95% confidence interval calculated by the model, at D28 a higher

**Table 3. Reasons of exclusion during the study.**

| Reasons for exclusion | Number of cats |
|---|---|
| Missing data at any point of the study | 140 |
| Not present for the required minimal time for observation | 58 |
| Diffuser definitively unplugged before D28 | 15 |
| Scratching post not available throughout study | 132 |
| Forbidden concomitant treatment | 3 |
| Environemental changes | 6 |
| Cat hospitalization | 1 |

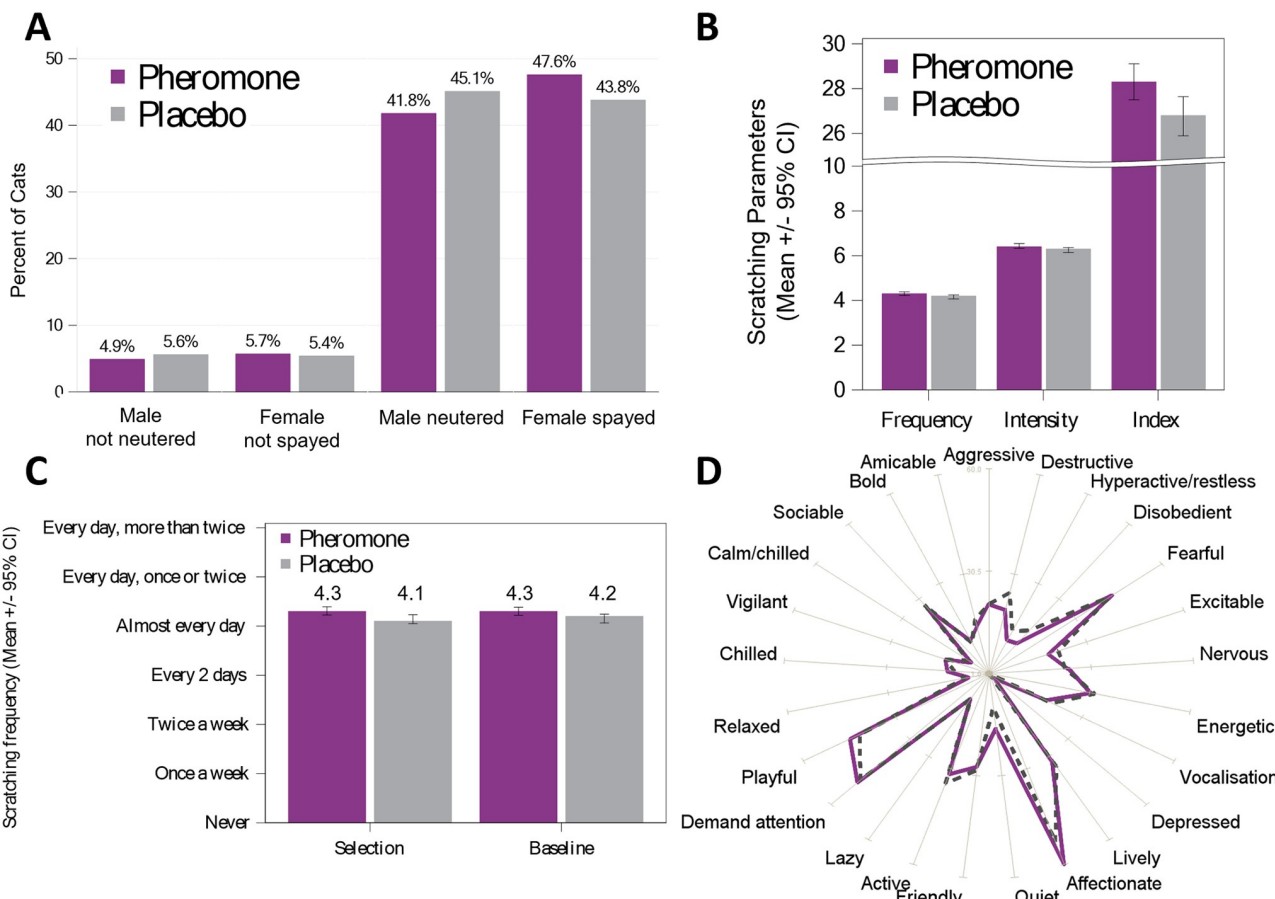

**Fig 2. Population characteristics at baseline.** A) Gender distribution. B) Data at baseline (D0) of the principal scratching parameters. C) Impact of the caregiver involvement on their evaluation of the Frequency of undesirable scratching between Selection and Baseline (D0). D) Mean cat personality profile per group (the purple line represents the Pheromone Group and the grey dashed line the Placebo Group).

probability to be in the category "never" was highlighted for the Pheromone Group (17.3%) compared to the Placebo Group (9.1%) whereas at D0 the probabilities to be in "never" were around 1% for both groups. At D28 there was also a higher probability (16.8%) for the Placebo Group to be in the category "7 and more per week" compared to the Pheromone Group (8.8%), while at D0 the probabilities were 49.0% for the Pheromone Group and 43.1% for the Placebo Group. These results demonstrate a decreased probability for cats to be in high scratching Frequency category after 28 days, and that there is an increased probability for the undesirable scratching to have stopped at D28, with this being more likely for cats in the Pheromone Group.

The Intensity change from baseline between groups was assessed with a longitudinal model. The day of the study (D7, D14 and D28) (p<0.001) and the group (p<0.001) were significant, as was their interaction (p = 0.002). The Intensity change from Baseline was significantly different between the Pheromone and Placebo groups at D7 (p = 0.0170), at D14 (p = 0.0189) and at D28 (p<0.001) (see Fig 3B). For each day, the change from baseline of the Intensity of scratching was bigger for the Pheromone than for Placebo.

The Global Index of undesirable scratching is presented in Fig 3C. This was analyzed again with a longitudinal mixed model with repeated measures where several covariables were tested

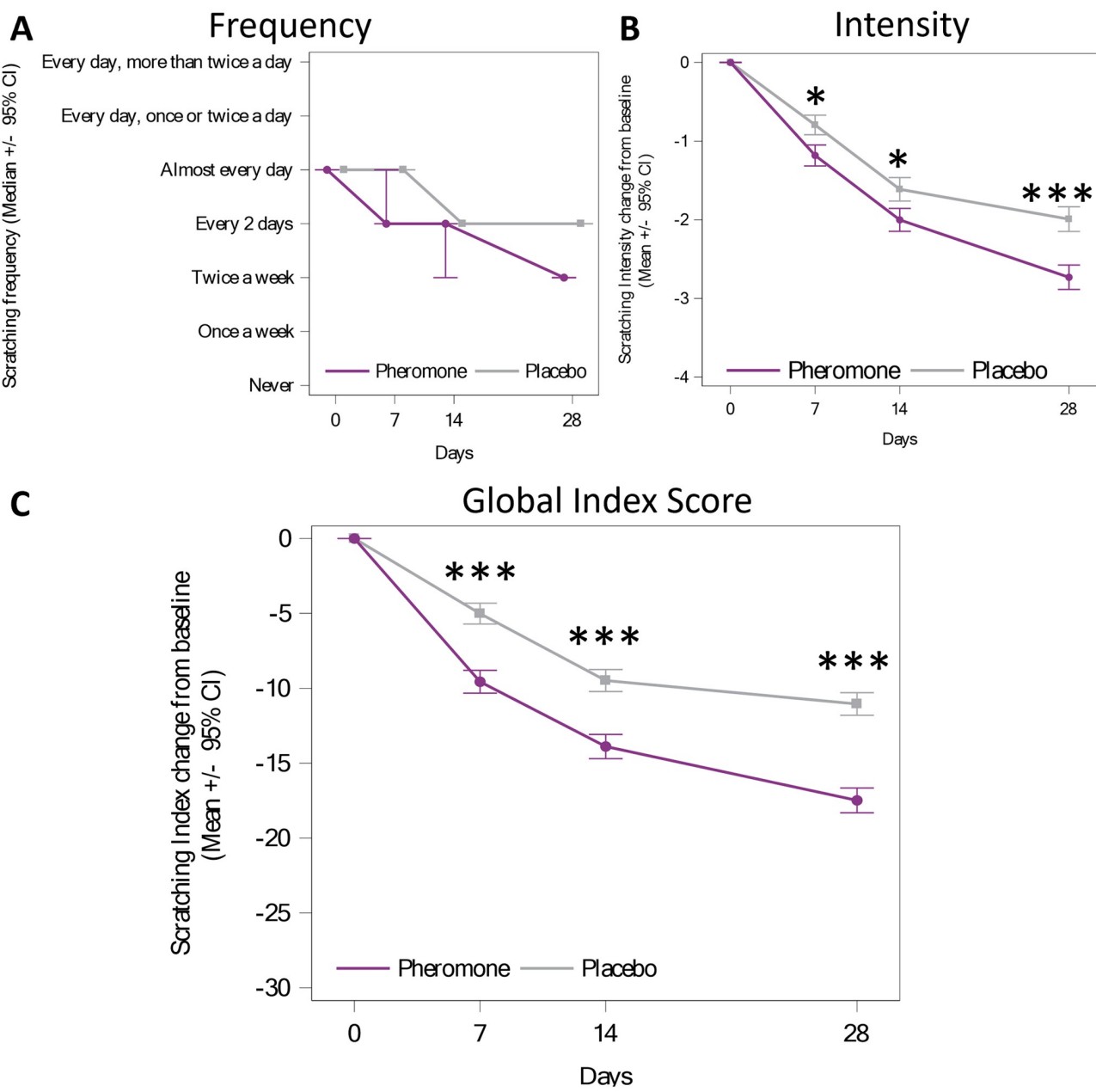

**Fig 3.** A) Change of the median Frequency for 28 days. B) Absolute change of the Intensity over 28 days. C) Change of the global undesirable scratching problem using the absolute change of the Global Index Score. ** represents significant difference of at least p<0.01 and *** for p<0.001.

(see statistical analysis section) and no interaction with product application was found. Again, a significant effect of the day (p<0.001), the group (p<0.001) and their interaction (p<0.001) was observed. The decrease of the Global Index Score is significantly higher for the Pheromone Group compared with the Placebo Group on days D7, D14, D28 (p<0.001).

No significant interaction between the covariables studied (see Table 2) and the group was found, meaning the effect of the pheromone on the scratching behaviour remained similar regardless of the initial conditions tested in this study.

**Table 4. Analysis of number of cats whose frequency, intensity and index score was reduced.**

| Variable | % (N) | FELIWAY® Classic | Placebo | P value (Xi 2) |
|---|---|---|---|---|
| Frequency | Cats that decreased by at least one category | 83.5% (456) | 68.5% (352) | **p<0.001** |
| | Cats that stopped undesirable scratching | 15.2% (83) | 12.8% (66) | p = 0.2690 |
| Intensity | Cats that decreased by at least 50% | 37.7% (206) | 27.8% (143) | **p<0.001** |
| Index | Cats that decreased by at least 50% | 68.1% (372) | 46.5% (239) | **p<0.001** |

The number of cats with a Frequency decrease of at least one category (not regrouped) at D28 is presented in Table 4. In addition, the number of cats with a reduction of at least 50% of the Intensity or Global Index Score is also documented in Table 4. For these three variables, this reduction was significantly higher for the Pheromone Group. Only the number of cats that stopped completely (meaning the Frequency was reported by caregivers to be 0) was not found to be significantly different (p = 0.269).

## Caregiver's perception

To assess the perception of the caregivers, throughout the study they were asked to evaluate how disturbing they considered the scratching over the last seven days using a visual analogue scale (see Fig 4A). A longitudinal mixed model with repeated measures was performed to explain the Disturbing level change from baseline, and the group had a significant effect with a bigger decrease in the Pheromone Group (p< 0.001). There was no interaction between the group and the day (p = 0.10).

Only 10.6% in the Pheromone group did not report any change either in Frequency or Intensity compared to 27.4% in the Placebo Group (p<0.001) (See Fig 4B).

A good correlation between the Disturbing value and the Global Index Score was found (R squared = 47% and p<0.001 with simple linear regression), however a large variability around the linear regression was observed (see S2 Fig).

## Safety

On the weekly questionnaire caregivers where asked to report any changes and adverse event. All adverse events and changes reported were included in the Fig 5. All declarations from caregivers were coded using the VeDDRA (Veterinary Dictionary for Drug Regulatory Activities) list of clinical terms and any behaviour changes reported by caregivers during the study that did not fit with a predetermined term were regrouped under "behaviour disorders" [29].

In this large study, more adverse events were observed in cats receiving the placebo (12.6%) than the pheromone (7.4%). More specifically, general behaviour disorders, urination outside of the litter box, lethargy and vocalisation (meowing) appeared more frequently in the Placebo Group than the Pheromone Group (see Fig 5).

## Discussion

Scratching plays a key role in social communication between cats as a marking behaviour by delivering visual and chemical messages that have an immediate and/or long-lasting impact on emotional states and behaviours [1–3]. Although undesirable scratching is one of the most frequent behaviour problems reported by cat owners, which may lead to relinquishment of the cats [1, 30], there is limited published research investigating the effect of using olfactory safety messages, i.e. the synthetic F3 fraction of the FFP to reduce this behaviour. This study provides direct evidence that the use of the synthetic F3 fraction of the FFP

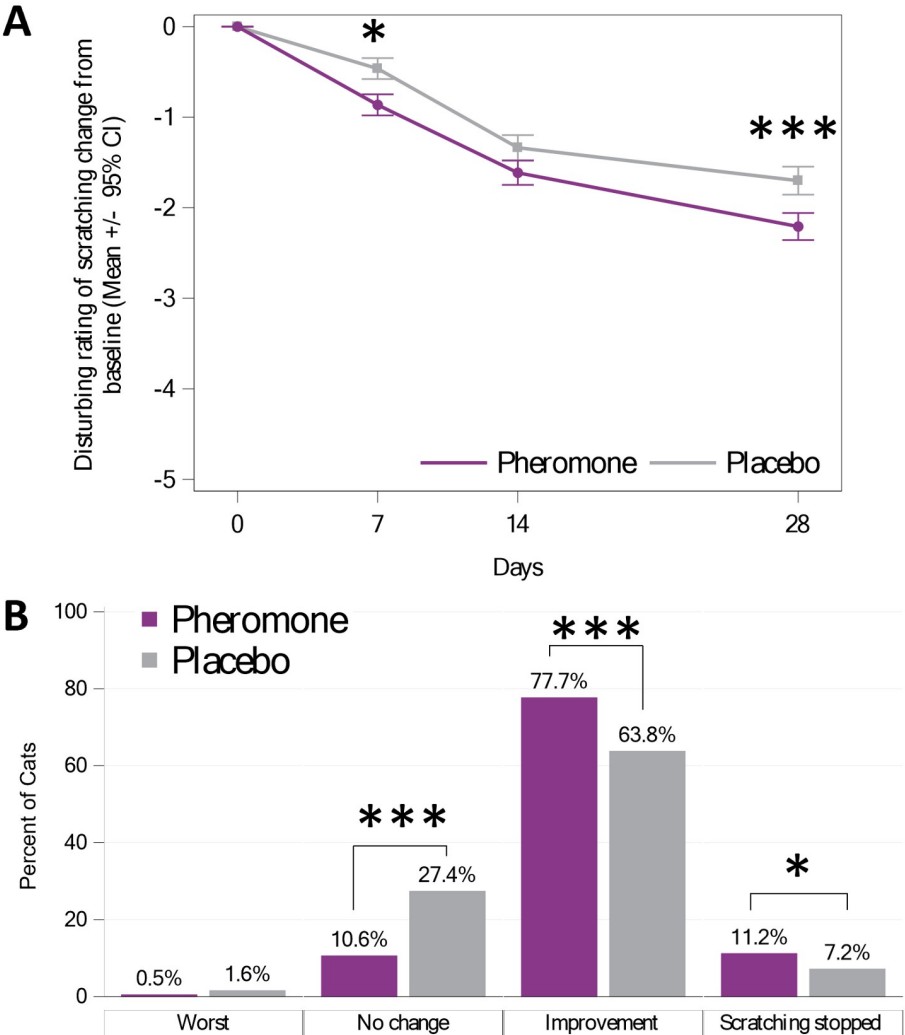

**Fig 4.** A) Change of the perceived disturbance due to scratching during the previous 7 days. B) Overall evaluation of the scratching problem over the last 28 days using Xi 2 test. * represents significant difference of at least $p < 0.05$ and *** for $p < 0.001$.

significantly reduces the Frequency, Intensity and Global Index Score of undesirable scratching using a placebo-controlled group. This is in keeping with the positive impact of facial pheromones on the emotional state and social communication in cats [5] and, further, is in line with recent studies suggesting the concomitant use of different pheromone products can reduce undesirable scratching in home environment [10, 11, 31]. The F3 fraction of the FFP possibly reduces the need for marking behaviour by providing olfactory messages for environmental safety [3–5].

Most of caregivers (88.9%, pooling "improvement" and "scratching stopped" in Fig 4) in the Pheromone Group reported a marked decrease in undesired scratching regardless of the type or size of scratching post or stress source—which showed a significant difference when compared to the Placebo Group (71%, pooling "improvement" and "scratching stopped" in Fig 4). These results suggest that the use of pheromone products is promising in terms of managing undesired scratching in home environments.

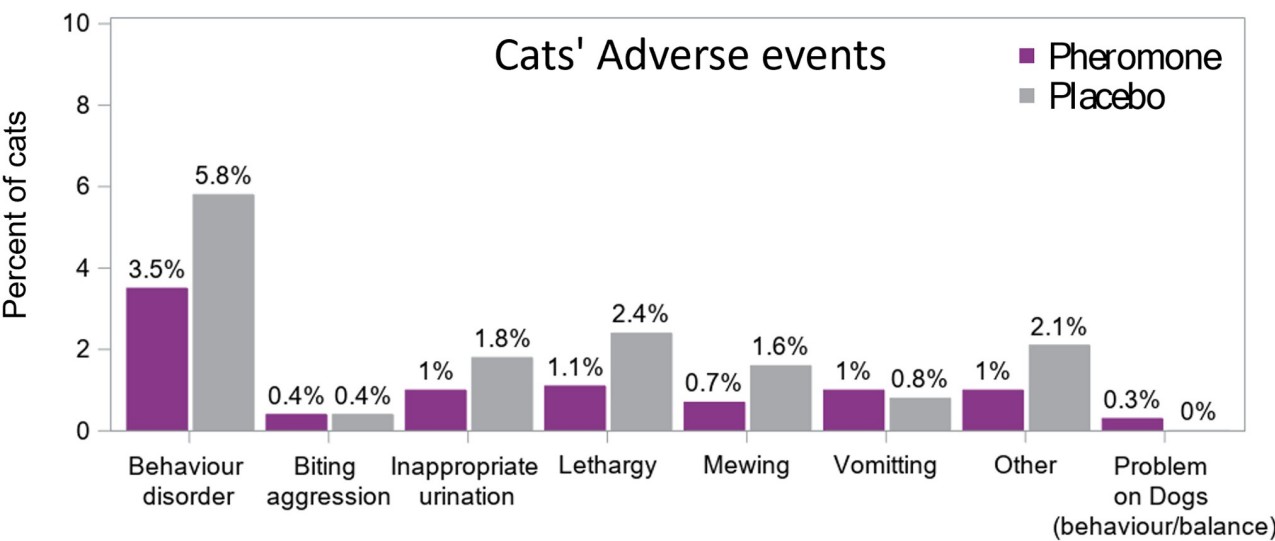

**Fig 5. Adverse events for cats classified according to VeDDRA terms.** Behaviour disorder includes a large variety of behaviour changes such as a cat's change in routine, etc...

Studies to date have demonstrated that cat scratching behaviour can be redirected to appropriate scratching posts provided in home environments [8, 11]. However, caregivers often use punishment and confrontational measures to stop this behaviour which only serves to increase scratch marking due to heightened stress and anxiety in cats [1]. Thus, even when the appropriate scratching posts are provided undesired scratching behaviour might continue to exist in the case of ongoing stress caused by social tension and/or an inability to meet environmental needs of cats [6–8, 10]. All the cats that participated in this study were reported to have at least one scratching post available but were still using home furnishings to scratch. The majority of the cats (69.4%) also used their post and most of them (71.4%) had their scratching posts in the same room where the unwanted behaviour occurred. Though scratching posts should be taller than the cat fully stretched to meet the cat's behavioural needs, this study found that only 45.8% of the caregivers in the study provided their cats with a scratching post that met these criteria. This might provide some insight into why cats are scratching their environment—more than half of the cats that participated in this study did not seem to have access to adequate means of expressing their natural scratching behaviour. This result supports the view that caregivers do not have sufficient knowledge about the behavioural needs of their cats [32]. Even though nearly half of the cats had scratching posts of the appropriate size, this did not prevent them to use the furniture for scratching, indicating that stress marking might be quite common.

Interestingly, the Frequency reported by the caregivers during the selection phase, based on what they could remember, was similar to the value declared at D0 after having the possibility to observe the cat more closely knowing more about the study. This observation is interesting since this may show that having a long baseline period allowing dedicated observation before starting an intervention might not be important. This also informs that undesirable scratching is a relatively stable behaviour.

Another interesting result of this study is that the majority of cat caregivers stated that their cat was unfriendly and hesitant about approaching strangers (see S1–S3 Figs). To our knowledge, this is the first demonstration of a possible link between cats' temperament and

unwanted scratching behaviour in the home environment. This supports the findings of a recent study suggesting that cats with a non-relaxed temperament were more likely to show unwanted elimination compared to cats with a relaxed temperament [33]. This also demonstrates the importance of olfactory safety for cats and explains why and how the F3 Fraction of the Feline Facial Pheromone is effective in reducing unwanted scratching.

On the 28th day of FELIWAY® Classic diffuser usage 11.2% of caregivers declared that the cats in the treatment group had completely stopped scratching in contrast to placebo group (7.2%). Interestingly, even when the scratching Frequency was reported as 0 during the previous week, the level of disturbance mean was still around a value of 2 (see S2 and S3 Figs). This suggests that the scale used to assess the potential disturbance will always be influenced by previous memories; improvements to the scale in future studies could avoid this. It is interesting to note the difference between the percentage of cat that stopped scratching the last 7 days (Frequency = never 0, in the Table 4) and the percent of owner declaring the cat has stopped scratching over the 28 days (in Fig 4B). A higher percentage of cats that stopped scratching was observed on the seven last days when compared to the global period of 28 days. This might be explained by the cats that stopped scratching closer to the last days (the 7 last days) and who have not been considered by the owner in the global question about the overall 28 days. For future studies, this question on the questionnaire might need to be rephrased to avoid this confusion.

In this study, any weekly change detected by the owner were reported without determining the possible link or not with the product. Any change in cat routine or behavior were reported event positive change for owner like the cat is calmer or cuddlier. For cats, the percentages of adverse effects for the Pheromone group were seemingly low (7.4%) and a higher percentage was observed for the Placebo Group (12.6%). These findings show that FELIWAY® Classic Diffuser is a safe product.

In addition, it has been reported that the majority of the cat owners (73.1%) indicated that they would recommend this product to individuals who were experiencing similar problems with their cats.

The fact that behaviours such as elimination outside of the litter tray, vocalization and other general behaviour changes were more frequently reported as adverse effects in the Placebo Group, this may be due to a global calming effect of the F3 fraction of FFP that helped decrease the Frequency of other disturbing behaviours, although a baseline comparison would be needed to confirm if these behaviours were already present before the study to better support a conclusion about this.

## Conclusion

This blinded, randomized and placebo-controlled study shows that the use of the synthetic version of FELIWAY® Classic Diffuser, reduces undesired environmental scratching of cats (both the Frequency and the Intensity) with 11.2% of the owners declaring a complete stop of scratching behaviour of their cats. This study also found that 71.4% of the cats would display undesired scratching even if they use a scratching post and that majority of cats were reportedly unfriendly and hesitate about approaching strangers maybe bringing to light a possible link between cats' temperament and unwanted scratching behaviour. This might indicate that cats could be scratching the furniture as an expression of stress and the pheromone product may reduce this stress related behavioural manifestation. The use of this simple and innocuous management approach, possibly in conjunction with environmental adjustment to the cat's needs could save cats from unpleasant procedures such as onychectomy, relinquishment, or euthanasia.

## Supporting information

**S1 Fig. Probability of each regrouped scratching modalities according days and groups.**
(TIF)

**S2 Fig. Correlation between the scratching index and the owner disturbing assessment.**
(TIF)

**S3 Fig. Bar plot of the mean of owner assessment of the disturbing level during the last assessment (D28) and they final global evaluation (D28) and the actual frequency reported present in 2 categories.**
(TIF)

**S1 File. This contain the exact value used for the different graphs and figure of this article.**
(PDF)

**S2 File. This contain the questionnaire used every week for this study to record scratching behaviour.**
(PDF)

**S3 File. Information on cat scratching device and cat behavior with visitor.** Cats profile at baseline (extract from Fig 2B).
(TIF)

## Acknowledgments

The authors acknowledge the cooperation of the caregivers that participated in this study. Authors would also like to thank Khnata Lamchemmachan and her team for organizing product logistics and transport.

## Author Contributions

**Conceptualization:** Joana Soares Pereira, Yasemin Salgirli Demirbas, Laurianne Meppiel, Sarah Endersby, Gonçalo da Graça Pereira, Xavier De Jaeger.

**Formal analysis:** Laurianne Meppiel.

**Methodology:** Joana Soares Pereira, Yasemin Salgirli Demirbas, Laurianne Meppiel, Sarah Endersby, Gonçalo da Graça Pereira, Xavier De Jaeger.

**Supervision:** Xavier De Jaeger.

**Validation:** Laurianne Meppiel.

**Writing – original draft:** Joana Soares Pereira, Yasemin Salgirli Demirbas, Gonçalo da Graça Pereira, Xavier De Jaeger.

**Writing – review & editing:** Joana Soares Pereira, Yasemin Salgirli Demirbas, Laurianne Meppiel, Sarah Endersby, Gonçalo da Graça Pereira, Xavier De Jaeger.

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
