## [Decision Letter · Decision Letter 0]

2 Jun 2023

PONE-D-23-09288The impact of the use of FELIWAY® Classic Diffuser in reducing undesirable scratching in catsPLOS ONE

Dear Dr. De Jaeger,

Thank you for submitting your manuscript to PLOS ONE. After careful consideration, we feel that it has merit but does not fully meet PLOS ONE’s publication criteria as it currently stands. Therefore, we invite you to submit a revised version of the manuscript that addresses the points raised during the review process. Although the review overall recognized the quality of your research they are looking for additional information in a few areas and expressed some concerns especially with regards to the flow and readability of your manuscript. 

We look forward to receiving your revised manuscript.

Kind regards,

Cord M. Brundage, D.V.M., Ph.D.

Academic Editor

PLOS ONE

2. You indicated that ethical approval was not necessary for your study. We understand that the framework for ethical oversight requirements for studies of this type may differ depending on the setting and we would appreciate some further clarification regarding your research. Could you please provide further details on why your study is exempt from the need for approval and confirmation from your institutional review board or research ethics committee (e.g., in the form of a letter or email correspondence) that ethics review was not necessary for this study? Please include a copy of the correspondence as an ""Other"" file.

“LM, SE and XDJ are employees of Ceva Santé Animale. 

JSP, YS, GGP authors received fees from Ceva Santé Animale for their contribution on the study design, interpretation of results and manuscript writing.  “

Reviewers' comments:

Reviewer's Responses to Questions

**Comments to the Author**

1. Is the manuscript technically sound, and do the data support the conclusions?

Reviewer #1: Yes

Reviewer #2: Partly

2. Has the statistical analysis been performed appropriately and rigorously? 

Reviewer #1: Yes

Reviewer #2: I Don't Know

3. Have the authors made all data underlying the findings in their manuscript fully available?

Reviewer #1: Yes

Reviewer #2: No

4. Is the manuscript presented in an intelligible fashion and written in standard English?

Reviewer #1: Yes

Reviewer #2: No

5. Review Comments to the Author

Reviewer #1: The current manuscript examines the efficacy of FELIWAY® Classic Diffuser in reducing undesirable scratching in cats.

I would suggest making the title more specific; perhaps you could change from “The impact of the use of FELIWAY® Classic Diffuser in reducing undesirable scratching in cats” to: “Efficacy of the Feliway ® Classic Diffuser in reducing undesirable scratching in cats: a randomised, triple-blind, placebo-controlled study”

I thoroughly enjoyed reading the manuscript and found it very interesting. I look forward to seeing

it published. My comments are only minor.

You have a nice abstract, but I would suggest mentioning a little bit as introduction the scratching behaviour (if word count allows).

The introduction is nice, clear, and well-written. It has a nice flow and clearly states the aims of the

study. Perhaps a little bit of extra detail on the conclusion may be useful. To manage scratching on

furniture I would like to know from a general overview of the recent and current studies if it were better to use F3, FIS or both in reducing scratching in terms of efficacy and costs.

The reference 3 is incomplete.

Reviewer #2: Overall the research approach seems sound, and this seems to be an important part of behavioral research. However the writing require significant revision both in terms of flow and readability but also in cohesion. There are many sections of information that are first mentioned in the results or even discussion instead of in the methods. Items reported in the text to imply significance are not always presented with p values and/or when they are the figure does not contain any visual indicators of significance such as asterisks. There is not sufficient information about what kind of regression/model was used or built for me to evaluate if it was done appropriately, and the same is true for some of the correlations.

ABSTRACT

Line 16: consider with instead of ‘that included’ especially considering line 18 also uses the word included.

Line 25- what does ‘one category’ mean? The reader doesn’t know enough at this point to understand that.

Line26: nice to have p values, please explicitly state which group had lower scratching at these days with is the key finding

Line 30: include the word diffuser after Classic

INTRODUCTION

Line 35- do we have a citation to support this sentence?

Line 37- ‘will’ is unnecessary.

Line 40-41, the sentence starting with ‘The use of scratching as a marking signal is expected’ seems out of place in this paragraph and may be more relevant later in the introduction. Consider shortening by removing ‘that’. Please also add a citation.

Line 51- citation 7 is really only relevant to onychectomy, please provide citations for the other recommendations

Line 51-56- this is awkward to have lumped into one sentence for the reader, please consider splitting and reworking (the phrase ‘a procedure’ is also unnecessary, it’s not clear what the different populations of cats are here).

Line 61- this would be a good place to introduce scratcher preferences, needs

Line 63- ‘being’ is unnecessary.

Line 63-67 is an awkwardly long sentence.

MATERIALS AND METHODS

Lines 88-92 this seems like the sort of calculation that might belong under statistical analysis. Also what kind of regression model?

There should be more information about recruiting here- what kind of panel of cat caregivers, how were they recruited other than emailed (were they sent the whole survey? Or asked to sign up?)

DATA COLLECTION

Line 110-112- please consider starting the paragraph with this sentence or put it in the design section. Please also include the window of opportunity allowed for each time point- 24 hours in other side, only on the day of? 3 days after the email notification?

Line 117: consider lumping all of the human participant information together (age, consent). Then you can separate out all the cat inclusion and exclusion criteria.

Line 120: please define 2 per week- 2 bouts, 2 minutes, 2 hours?

Line 121: observed (aka the client saw the cat doing the damage) or found (during or after?)

Line 123- likert scales can be treated statistically as semi-quantitative scales, but this should be supported with a citation. Also consider changing ‘every day’ to ‘daily’ in the text for brevity and ease of reading, even if that is not what the survey said.

Line 125-126 Were examples of intensity provided or was it entirely arbitrary?

Line 128- tense, please switch to past tense

Line 131- “from these two criteria’ is redundant

Line 133: “in addition to this is unnecessary”. Please confirm the clients were asked to rate disturbing on day 0 here, as only day 28 is mentioned. Consider a table or using different demarcation system than parentheticals inside parentheticals.

Somewhere in this section, the temperament/personality questions need to be introduced instead of in the statistical analysis.

EXCLUSION CRITERIA- could consider having this only be a table and earlier in the method section. Please define “Major environmental change” is that a new piece of furniture, a move, some one leaving the house hold, all of the above?

Is there a citation/justification for needing the client to observe 4/hrs a day?

You need to include having a data point missing as exclusion criteria since it later showed up as a reason for exclusion (table 3)

Line 15-152- this sentence does not belong in the statistical analysis section, that’s data collection. The sentence on the Global Index Score is redundant to the previous section.

Line 153-155- you state it’s not appropriate to report significance, but then later do so, and most scientific readers will want to know anyway if the two groups were different, even if we know it was just due to chance, because it changes our interpretation of the comparisons later.

Line 161- please justify why you wanted to limited the number of categories as you lose data and variability when doing this- small cells?

Line 165- evolution is not an accurate term here, we are not watching changes over generations, please revise all instances of this verb.

Line 166- why was Chi squared sused for the Global Index score? It was the closest to truly continuous variable you had. It’s statistically appropriate to do so (not violating assumptions) but not a choice many would make and would love to see some discussion on this.

Line 167- this sentence is unclear because of ‘the same analyses were performed’. If you mean as each other, you can remove that phrase and join ‘for intensity and global index score’ a XXXXX was used. However, in the sentence just above it, you state that Chi squared was used.

Line169- what kind of longitudinal model (linear, survival?) and how was it built? ‘ per behaviour observation’ could be shortened to ‘by’ to make the sentence easier to read.

Line 173- do you mean generalized least square means?

RESULTS

Line 183- ‘for different reasons’ is unnecessary just refer to the table.

Line 191- this is where you contradict lines 153-155

Line 194- first time the phrase ‘selection phase’ is mentioned- please define in methods section and in diagram.

Line 197- first time the cat personality profile is mentioned other than abstract, include in methods. Please also expand on this section is this a validated scale?

Line 204: Figure 2 caption- this is not particularly clear if you are displaying frequency and intensity of scratching, label it that instead of ‘data’ Item c- caregiver involvement is an entirely new variable that did not appear anywhere in the method section.

3.2 EFFICACY of F3

Line 212- I think you mean “non-continuous”? please remove unnecessary words for clarity- the, parameter, of the scratching frequency. In, not on.

Line 213- which quantitative test? And why ‘but’ the ordinal regression model was used? This should also be discussed in the statistical analysis section.

Line 215-221: Your figure has no asterisks suggesting visually that none of these comparisons you are listing were statistically significantly different.. no p values are reported in this section.

Line 221-23- this type of evaluative sentence goes in the discussion, numbers only for the results.

Lines 224-228: please tell us a little bit about the direction of these differences and interactions.

Line 229: shorten to ‘ The Global Score’ is presented in Figure 3C.

Line 230- again which kind of mixed model? With what variables as repeated/between/predictors? This information isn’t in the supplementary material either

Line 231- would stick with calling this variable ‘day’ vs study day, please also here expand a little on the interaction. Most models I’ve seen presented will also provide table with the intercepts, effect size and variables.

Line 237- typically the asterisk and significance goes in the caption vs the figure title.

Line 240- please list the variables because it’s not clear from your materials and methods and/or say ‘no other significant interaction was found. The second part of this ‘meaning that…’ goes in the discussion.

3.3 Care giver perception

Lines 253-255 do not belong in results

Line 255- again which kind of longitudinal model?. Consider splitting into two sentences with the lack of interaction it own sentence.

Line 258- don’t need the introductory phrase ‘concerning the care giber’s perception of the scratching problem’

Line 259-260 I think you mean report? Is there a Chi squared and p value to go with this?

Line 261-262: what kind of correlation? R squared, rho? Please report the p value and regression type for this correlation.

Line 265- this is the first time disturbingness is mentioned that it’s limited to prior 7 days, please add this to the method section. Shortened line 266 to “evaluation of scratching over last 28 days using chi squared”

Line 270- please defined VeDDRA as this is it’s first time being mentioned.

Line 274- please include in the method section that adverse events were recorded. Please report a p value for the comparison between pheromone and placebo adverse events. There are again no asterisks in the associated figure

Line 277- this interpretive sentence goes in the discussion.

DISCUSSION:

Line 288- consider removing ‘only’ and ‘on’ for improved clarity and flow

Line 289- introduce the idea of safety signals earlier or define here. This is the first time you have used the abbreviation FFP, please introduce earlier or define or both.

Line291-292- how is at home with clients a controlled setting? Do you mean placebo controlled?

Line 292- consider ‘in keeping with’ instead of “is not surprising considering”

Line 294- remove ‘that’ for easier reading.

Line 296: remove ‘of cats for easier reading. Would like a citation for this statement.

Line 297- ‘considering that’ is unnecessary, removing ‘in this study’ would increase ease of reading, would be best to compare to the placebo group and end the sentence at source. Separate line 300 on into it’s own sentence or use a semi colon.

Line 301- I contend that if you have to lump improved with stopped completely this is a ‘promising’ product, not a ‘highly promising’ product.

Line 303 “ however’ will be shorter to ready than ‘on the other hand’ you also did not quite present an ‘on the one hand’.

Line 305- please define what ‘’correct use’ of scratching post means here- do you mean providing appropriate posts in appropriate location by the client, or do you mean scratching the provided posts by the cat?

Line 308- sub ‘had’ for were reported to have’ for ease of reading.

Line 310- had, (past tense not have)

Line 311- might be clearer to say taller or longer and length vs size of the cat.

Line 312-316 do you mean your study, or the one that the previously cited one (4,5,720). If this is from you study, you need to include information about the cats scratching posts from the recruitment phase in your methods and in your results if you want to include it in your discussion. Please also convert all to past tense. “more than half the cats that participated in the study did not seme to” is redundant to the percentage presented above. This sentence also is too long and should be reworked into multiple sentences.

Line 318- consider starting at Even though. What about location of post?

Line 321-322- consider reported vs declared, and ‘based on memory’ and ‘after potentially observing the cat more closely because of the study” for better clarity.

Line 234- the parenthetical is unnecessary.

Line 326- double check with editor if personal pronouns are allowed often they are not.

Line 328- use past tense.

Line 329-330- split into two sentences for easier reading.

Line 332- this sentence actually seems unrelated to the rest of the paragraph- please move somewhere else or remove entirely.

Line 335: consider ‘caregivers reported cats had completely stopped scratching in contrast to the placebo group (7.2%).’

Line 338- remove ‘was still considered with a’ and place ‘was still’ after the word ‘mean’

Line 339- please use suggests or indicates instead of ‘means’ since Mean was just used to refer to a number. Consider reworking this sentence to explain that previous memories influence reporting on disturbingness vs calling it a residual.

Line 341- easier to read as ‘it is interesting to note the difference…”

Line 344: higher percentage of what?

Line 345- convert to past tense, do you mean could not have been counted by the owner?

Line 348- this is a REALLY high number of adverse events for a pheromone based on clinical experience and generally held belief that you cannot overdose on this product. Please report and discuss how this compares to other Feliway Studies. Please also discuss nocebo and placebo effects

CONCLUSION

Line 363- shorten to “Feliway Classic Diffuser reduces frequency and intensity of undesired scratching”

Lines 365- 367- the stress derivation of scratching was your a priori assumption, and was not tested here so you can’t say this paper supports cats scratching furniture may be in poor welfare state. Rework this please.

Line 367- start the sentence at ‘the use of this simple and innocuous’ Also please discuss the side effects rates in other papers before declaring this to be innocuous.

Line 369= be explicit about unpleasant procedures- nail caps? Onchyectomy? This won’t decrease the need for say dental cleanings which may also be unpleasant for cats.

FIGURE1- the ‘recruiting phase’ is referenced several times in the paper but does not appear in the diagram. The font condensed (bold with small spaces between letters) and does not read well if printed full page- it looks like the text was converted to image for the red and blue text, consider revising- also please double check if this color scheme is red-green color blind compatible. I did not want to upload the images to an online tester to breach confidentiality.

FIGURE2- B: instead of calling it efficacy parameters which doesn’t inherently mean anything, can the y axis be titled ‘scratching parameters’?

C: can’t read the 3s over the dark purple very well- be consistent, the other graph showing numbers has them above the barks.

D: very hard to read the grey over the purple= can this be expanded?

FIGURE3: all: Please use different indicators for the points such as a dot and a square or a dot and a diamond vs only relying on the color.

A: Please make the error bars different color or line thickness form the placebo line- why are they're no error bars on day 28?

C: is this the global score index? Be consistent in your variable names. Again please also use different dogs

FIGURE 4: could call this ‘Disturbing rating’ for ease of reading and shorting in the axis labels. {Lease use different indictor shapes for the points for each group.

FIGURE 5: no asterisks suggests these are all statistically the same rate between placebo and Pheromone.

TABLE 2. This table is exceedingly hard to read. Please revise with each variable on one line in one column, then a row of the levels in that variable to it’s right:

Age 7m-2 year

3-6 year

7-10 year

Lifestyle Etc.

TABLE 4: replace all ‘of’s with ‘by‘

SUPPLEMENTARY MATERAIL:

Please use different point indicator for placebo vs pheromone.

Fig 1: no asterisks, so these are all not statistically significant?

6. PLOS authors have the option to publish the peer review history of their article (what does this mean?). If published, this will include your full peer review and any attached files.

Reviewer #1: No

Reviewer #2: No

---

## [Author Response · Author response to Decision Letter 0]

9 Aug 2023

General comments:

1) We hope that the modification made on the revised manuscript will cover the reviewer expectation about the data and the conclusion.

2) We added several information concerning the statistic to be clarify what was made

3) In the supportive information the minimal data set were available as describe in the PLOS data policy. The values used to build graphs are fully available and the questionnaire used are also available.

4) The modification made on the revised manuscript should help the manuscript to be more intelligible.

5) See our response to the reviewers in the documents.

---

## [Decision Letter · Decision Letter 1]

5 Sep 2023

PONE-D-23-09288R1Efficacy of the Feliway ® Classic Diffuser in reducing undesirable scratching in cats: a randomised, triple-blind, placebo-controlled studyPLOS ONE

Dear Dr. De Jaeger,

Thank you for submitting your manuscript to PLOS ONE. After careful consideration, we feel that it has merit but does not fully meet PLOS ONE’s publication criteria as it currently stands. Therefore, we invite you to submit a revised version of the manuscript that addresses the points raised during the review process. Reviewer 2 had only a few points that require clarification (e.g. line 294) and a few editing revisions that will hopefully be quick to resolve for manuscript acceptance.

We look forward to receiving your revised manuscript.

Kind regards,

Cord M. Brundage, D.V.M., Ph.D.

Academic Editor

PLOS ONE

Journal Requirements:

Reviewers' comments:

Reviewer's Responses to Questions

**Comments to the Author**

1. If the authors have adequately addressed your comments raised in a previous round of review and you feel that this manuscript is now acceptable for publication, you may indicate that here to bypass the “Comments to the Author” section, enter your conflict of interest statement in the “Confidential to Editor” section, and submit your "Accept" recommendation.

Reviewer #1: (No Response)

Reviewer #2: (No Response)

2. Is the manuscript technically sound, and do the data support the conclusions?

Reviewer #1: (No Response)

Reviewer #2: Partly

3. Has the statistical analysis been performed appropriately and rigorously? 

Reviewer #1: (No Response)

Reviewer #2: Yes

4. Have the authors made all data underlying the findings in their manuscript fully available?

Reviewer #1: (No Response)

Reviewer #2: Yes

5. Is the manuscript presented in an intelligible fashion and written in standard English?

Reviewer #1: (No Response)

Reviewer #2: Yes

6. Review Comments to the Author

Reviewer #1: (No Response)

Reviewer #2: Thank you for your time in approaching our comments. This is a significantly easier to read and follow paper from a process standpoint now as well.

A few edits for minor things. There is still a set of conclusions in the discussion that does not match the figures. It may be that the figures to be edited to match them as this is what the average reader will check first rather than the re-read the text of results. (when printing the figure on global evaluation of scratching at d28 which is all blue- has no figure label my apologies. This figure has no asterisk denoting significance. I would be surprised if 0 vs 199 was not significantly different distribution for "no change" and it seems like significant differnces are discussed in the results as well?

Line by line comments:

Line 51: un capitalize Frequency, since you are not yet discussing your data variable.

Line 67: replace the commas with decimal points in 0,3 ad 0,9m

Line 70: would prefer to have only and (vs and/or), as the implications that cats need more than 1 scratcher, which is typically the case

Line 73-76: may still benefit from being broken up into two sentences.

Line 77: again not discussing your data set so decapitalized Frequency

Line 97: rephrase with the use of the word “selection” only once

Line 115: change to reminder, insert ‘out’ after fill

Lines 117-129- thank you for expanding this and using bullets it is so much easier to follow along with now! Linw 128- the word ‘on’ is erroneous

Table 1- If one questionnaire was not answered

Line 294: how was “other cats at home” included in the model if only single cat households were used?

Line 206: can we expand on what ‘owner’s family situation’ means by referencing Table 2? (I first went digging in the supplementary material.

Thank you this is a much more in depth, repeatable statistical description!

Line 296: for consistency consider capitalizing Disturbing as a variable through out the paper

Line 337: this sentence is a little misleading compared to the figures and stats. Most clients in both groups reported improvement and it was only significantly different in the scratching stopped group per figure 4 B. There are no asterisk indicating significant differences for frequency in Figure 3A…

Line 349 and 350: : replace comma sin your percentages with decimal points

Line 410: consider “majority of cats were reportedly unfriendly and hesitate about approaching strangers//” for easier reading.

7. PLOS authors have the option to publish the peer review history of their article (what does this mean?). If published, this will include your full peer review and any attached files.

Reviewer #1: No

Reviewer #2: No

---

## [Author Response · Author response to Decision Letter 1]

8 Sep 2023

The details of our answer point by point are availalble in the document "Response to reviewer 2".

We hope that this will cover all aspect of

---

## [Editor Report · Decision Letter 2]

14 Sep 2023

Efficacy of the Feliway ® Classic Diffuser in reducing undesirable scratching in cats: a randomised, triple-blind, placebo-controlled study

PONE-D-23-09288R2

Dear Dr. De Jaeger,

We’re pleased to inform you that your manuscript has been judged scientifically suitable for publication and will be formally accepted for publication once it meets all outstanding technical requirements.

Kind regards,

Cord M. Brundage, D.V.M., Ph.D.

Academic Editor

PLOS ONE

---

## [Editor Report · Acceptance letter]

25 Sep 2023

PONE-D-23-09288R2 

Efficacy of the Feliway ® Classic Diffuser in reducing undesirable scratching in cats: a randomised, triple-blind, placebo-controlled study. 

Dear Dr. De Jaeger:

I'm pleased to inform you that your manuscript has been deemed suitable for publication in PLOS ONE. Congratulations! Your manuscript is now with our production department. 

Kind regards, 

on behalf of

Dr. Cord M. Brundage 

Academic Editor

PLOS ONE